# In Vivo Quantitative Imaging of Glioma Heterogeneity Employing Positron Emission Tomography

**DOI:** 10.3390/cancers14133139

**Published:** 2022-06-27

**Authors:** Cristina Barca, Claudia Foray, Bastian Zinnhardt, Alexandra Winkeler, Ulrich Herrlinger, Oliver M. Grauer, Andreas H. Jacobs

**Affiliations:** 1European Institute for Molecular Imaging (EIMI), University of Münster, D-48149 Münster, Germany; claudia.foray@uni-muenster.de (C.F.); bastian.zinnhardt@roche.com (B.Z.); 2Biomarkers & Translational Technologies (BTT), Pharma Research & Early Development (pRED), F. Hoffmann-La Roche Ltd., CH-4070 Basel, Switzerland; 3Université Paris-Saclay, CEA, CNRS, Inserm, BioMaps, Service Hospitalier Frédéric Joliot, F-91401 Orsay, France; alexandra.winkeler@cea.fr; 4Division of Clinical Neuro-Oncology, Department of Neurology, University Hospital Bonn, D-53105 Bonn, Germany; ulrich.herrlinger@ukbonn.de; 5Centre of Integrated Oncology (CIO), University Hospital Bonn, D-53127 Bonn, Germany; 6Department of Neurology with Institute of Translational Neurology, University Hospital Münster, D-48149 Münster, Germany; olivermartin.grauer@ukmuenster.de; 7Department of Geriatrics with Neurology, Johanniter Hospital, D-53113 Bonn, Germany

**Keywords:** glioblastoma, positron emission tomography, magnetic resonance imaging, translocator protein, tumor microenvironment, heterogeneity, [^18^F]FET, [^18^F]DPA-714

## Abstract

**Simple Summary:**

Defining glioma heterogeneity represents a promising strategy to unravel the mechanisms behind therapy resistance and tumor recurrence. The current review provides a comprehensive overview of experimental and clinical data concerning the visualization and quantification of the tumor microenvironment heterogeneity using molecular imaging, with a special emphasis on positron emission tomography (PET).

**Abstract:**

Glioblastoma is the most common primary brain tumor, highly aggressive by being proliferative, neovascularized and invasive, heavily infiltrated by immunosuppressive glioma-associated myeloid cells (GAMs), including glioma-associated microglia/macrophages (GAMM) and myeloid-derived suppressor cells (MDSCs). Quantifying GAMs by molecular imaging could support patient selection for GAMs-targeting immunotherapy, drug target engagement and further assessment of clinical response. Magnetic resonance imaging (MRI) and amino acid positron emission tomography (PET) are clinically established imaging methods informing on tumor size, localization and secondary phenomena but remain quite limited in defining tumor heterogeneity, a key feature of glioma resistance mechanisms. The combination of different imaging modalities improved the in vivo characterization of the tumor mass by defining functionally distinct tissues probably linked to tumor regression, progression and infiltration. In-depth image validation on tracer specificity, biological function and quantification is critical for clinical decision making. The current review provides a comprehensive overview of the relevant experimental and clinical data concerning the spatiotemporal relationship between tumor cells and GAMs using PET imaging, with a special interest in the combination of amino acid and translocator protein (TSPO) PET imaging to define heterogeneity and as therapy readouts.

## 1. Introduction

Glioblastoma (GBM) is a highly proliferative, invasive and neovascularized heterogeneous tumor tissue driven by an immunosuppressive tumor microenvironment (TME). Current therapies for GBM rely on surgical resection, radiotherapy and chemotherapy but patient prognosis remains limited. Therapy response depends on multiple factors, including intra- and inter-tumoral genetic, epigenetic, molecular and metabolic heterogeneity. In addition, radiotherapy and chemotherapy increase the necrotic core with underlying inflammation, vascular changes and blood-brain barrier disruption, leading to enhanced immune cell infiltration and brain edema. Accordingly, subsequent molecular and cellular changes limit the efficiency of the current therapies. In addition, therapy resistance is also established by the synergistic communication between glioma cells and their microenvironment. The secretion of inflammatory mediators and angiogenic factors by the tumor and immune cells promotes a pro-tumorigenic immunosuppressive environment, synergistically promoting tumor cell proliferation and immune cell infiltration within the TME. Furthermore, several immune checkpoint systems implemented by tumor cells also prevent the anti-tumorigenic activity of infiltrating lymphocytes.

Therefore, non-invasive imaging techniques, such as magnetic resonance imaging (MRI) and positron emission tomography (PET) could unravel and longitudinally capture the heterogeneity and spatio-temporal dynamics of the glioblastoma microenvironment in vivo, ultimately supporting the development of new immunotherapies targeting the different components of the TME [1].

## 2. The Tumor Microenvironment (TME)

Single-cell profiling of myeloid cells in glioblastoma across species and disease stages reveals macrophage competition and specialization over time [2]. In non-treated human GBM, the largest fraction of the immune cell population was constituted of tumor-associated myeloid cells (TAMCs) (82–97%), including tumor-associated microglia/macrophages (TAMs) and myeloid-derived suppressor cells (MDSCs), followed by T lymphocytes (20%). TAMs consist of two main cell populations differentiated by their ontogeny, microglia and myeloid-derived macrophages. Interestingly, recurrent human GBMs showed a more diverse immune compartment, with an increased fraction of lymphocytes, including T cells, natural killer (NK) and B cells. Similarly, in an experimental GL261 glioma mouse model, TAMs represented the largest immune cell fraction, divided into microglia and myeloid-derived macrophages, as observed in humans, with microglia outnumbering peripheral macrophages at the early stages. Microglia-derived TAMs were predominant in newly diagnosed gliomas but were outnumbered by monocyte-derived TAMs following recurrence, especially in a highly hypoxic tumor microenvironment [2]. Therefore, tracking the spatiotemporal dynamics of the resident and infiltrating immune cells could ultimately endorse prognosis.

Like TAMs, myeloid-derived suppressor cells (MDSCs) are a heterogeneous group of cells originating from myeloid precursor cells. MSDCs can be subdivided into polymorphonuclear (PMN)-MDSCs and monocytic (M)-MDSCs. PMN-MDSCs, regarded as pathologically activated neutrophils, are the main population of MDSCs in mice and humans [3]. As for TAMs, MDSCs create a pro-tumorigenic microenvironment by (i) reducing tumor infiltration of cytotoxic T cells, (ii) suppressing T cell function, (iii) promoting regulatory T cell (Treg) expansion, and (iv) promoting angiogenesis and invasion [4,5,6,7]. In accordance, an increased number of MDSCs is associated with a poor outcome in glioma patients [8]. Despite sharing common characteristics, TAMs and MDSCs can be distinguished based on their differential marker profiles and their temporal contribution to tumor initiation and progression [4].

To better understand the role and the dynamics of glioma-associated myeloid cells (GAMs) and other immune players on tumor progression, therapy response and recurrence, the development of new molecular imaging approaches is crucial to longitudinally characterize the immune components of the TME (Figure 1). Moreover, the development of immunotherapies and other therapeutic approaches requires non-invasive in-depth characterization of GBM and the establishment of new biomarkers to assess the TME spatiotemporal dynamics, as well as detection of a potential therapy-induced switch from an invasive to an anti-tumorigenic phenotype. Among the new imaging biomarkers, radioligands targeting the myeloid cell compartment are currently being investigated [9].

## 3. Quantitative Imaging of Glioblastoma

The contribution of non-invasive multimodal imaging to diagnosis and prognosis is increasingly being recognized. Clinical practice guidelines and recommendations in the management of suspected GBM comprise MRI with both T_2_-weighted (T_2_w), FLAIR and pre- and post-gadolinium enhanced T_1_-weighted (T_1_w) imaging. Diagnosis specificity could be improved by diffusion- and perfusion-weighted imaging to distinguish GBM from other tumor types and by nuclear imaging, such as amino acid PET imaging [11] (Table 1).

### 3.1. Magnetic Resonance Imaging

Diagnosis and treatment response are evaluated using contrast-enhanced T_1_w, T_2_w-MRI and FLAIR. Those MR sequences provide information not only on tumor size and localization but can also inform on secondary phenomena, such as the disruption of the blood-brain barrier (BBB), edema formation, hemorrhage and necrosis. Contrast-enhanced T_1_w-MRI has served as a surrogate for the highly aggressive part of the GBM, while its primary interpretation suggests BBB impairment rather than tumor growth. Besides, a substantial part of the tumor mass might not present dysregulated BBB, and, therefore, lack signal enhancement, biasing the tumor grade. Furthermore, MR imaging remains quite limited in discriminating metabolically distinct tissues within the tumor mass and, therefore, hampering tumor heterogeneity. Pseudoprogression, for example, a phenomenon likely related to therapy-induced changes in inflammation and permeability of the blood-brain barrier, is difficult to distinguish from true disease progression. Patients may first show an increase in gadolinium-enhanced glioma volume due to the anti-tumour-mediated immune response and localized inflammation, which may or may not be interpreted as tumor resistance or recurrence. However, the timing of T_1_w- and T_2_w-MRI changes (perfusion-weighted imaging) could still help to detect pseudoprogression [12]. Altogether, this phenomenon represents a crucial challenge for tumor therapy response and the MRI-based assessment of treatment planning.

Some of the inherent limitations of MR imaging can be overcome by using extended image analysis. Radiomics rely on the high-throughput extraction workflow of the advanced quantitative morphologic and textural features derived from static and dynamic images to unravel morphologic and textural patterns that could guide cancer management. In recent applications, radiomics could predict glioma grading with high accuracy (90%), proteomic, genomic and transcriptomic characteristics [13,14]. Moreover, new in the field, radiogenomics helps to resolve the genetically distinct subpopulations that coexist within the tumor microenvironment [15]. As a first step into the development of radiomics using PET imaging data, the [^18^F]FDOPA dynamics were found to predict the isocitrate dehydrogenase (*IDH*) mutations (a hallmark of CNS grade 2 or 3 astrocytomas) and the 1p/19q codeletion (a hallmark of CNS grade 2 or 3 oligodendrogliomas), which are determinant in patients’ prognosis [16,17].

Overall, neuroimaging using MRI offers limited insight into tumor heterogeneity and tissue differentiation, while emerging analyses try to overcome those limitations. Molecular imaging of amino acid transport as a surrogate marker for neovascularization and protein synthesis is one of the established clinical imaging biomarkers that could complement MRI-based diagnosis, tumor grading and delineation and treatment monitoring.

### 3.2. [^18^F]FET and Other Amino Acid PET Ligands

In highly proliferative brain tumors, the activity of neoplastic cells results in increased amino acid transport, supporting the use of radiolabeled amino acids as a biomarker for tumor growth. Among them, [^18^F]FET, [^18^F]FLT, [^11^C]MET and [^18^F]FDOPA were clinically tested in glioma patients [11]. [^18^F]FET is a valuable marker for active glioma volume with significantly higher sensitivity and specificity in the diagnosis of brain tumors over MR, [^18^F]FDG or [^18^F]FLT PET imaging [18,19,20,21], while these data still need further characterization. [^18^F]FET or [^11^C]MET signal exceeded the hyperintense glioma area depicted by T_2_w- or T_1_w-Gd-MRI and reported valuable complementary information to conventional MR images [21,22,23]. In post-therapy patients, the difference between FLAIR/T_2_w and [^11^C]MET hyperintense signal may delineate therapy-related tissue change, therefore, allowing the distinction between metabolic active tumor tissue and treatment-related changes in patients with gliomas [21].

A recent retrospective study analyzed 45 glioma patients treated with chemotherapy that underwent [^18^F]FET to discriminate between glioma recurrence and treatment-induced changes. The results indicated that [^18^F]FET PET imaging provided a good diagnostic performance (sensitivity: 86.2% (95% CI: 68.3–96.1%) and specificity 81.3% (95% CI: 54.4–96.1%)) [24], in line with another recent clinical study [25], while no gold standard for actual tumor progression was used. Interestingly, these studies also highlighted that [^18^F]FET accuracy in diagnosis was reduced in patients with *IDH*-mutant gliomas. Besides, metabolic changes after bevacizumab treatment could be identified by [^18^F]FET PET imaging earlier than structural changes detected by MRI, providing an earlier indication of glioma progression [20].

Similarly, [^18^F]FET uptake and volume were significantly reduced after adjuvant temozolomide (TMZ) chemotherapy in an experimental glioma model, in line with the concomitant decrease in gadolinium-enhanced T_1_w-MR-based tumor volume [26]. Likewise, Ceccon et al. (2021) indicated that TMZ induced a reduction in [^18^F]FET metabolic tumor volume (MTV) and maximum tumor-to-background ratio (TBR_max_) in 41 newly diagnosed glioma patients that underwent resection and two cycles of chemotherapy [27]. A decrease in MTV and TBR_max_ compared to baseline predicted a significantly longer progression-free survival (PFS) and overall survival (OS), respectively, independent of the methylation patient status. In those cases, MRI parameters did not show any significant change after therapy, highlighting the potential of [^18^F]FET PET imaging in the assessment of the treatment response compared to MRI. Overall, [^18^F]FET imaging could help patients’ management, including the diagnosis of pseudoprogression after TMZ chemotherapy and assessment of other treatment options.

Combining imaging modalities has become an essential tool for the assessment of tumor biology and heterogeneity in glioblastoma patients. Furthermore, defining heterogeneity using established biomarkers derived from different imaging modalities and investigating their spatiotemporal relationship has become a major approach to state the clinical condition and patients’ therapy response [11].

### 3.3. Translocator Protein (TSPO) PET Imaging

Within the heterogeneous glioma tissue, neoplastic tumor cells, glioma-associated microglia/macrophages, astrocytes, monocytic-MDSCs, endothelial cells and pericytes express the 18 kDa translocator protein (TSPO) [28,29]. TSPO ligands have been widely used as a marker for glioma and glioma-associated inflammation to decipher tumor heterogeneity of special interest in the field of targeted immunotherapies. TSPO has a pivotal role in tumorigenesis and glioma progression [30]; different studies demonstrated a positive correlation between TSPO expression and grade of malignancy in experimental glioma and human biopsies [31,32,33].

In recent years, a limited number of clinical studies (Table 2) has evaluated different TSPO PET tracers to measure glioma-associated inflammation in vivo. In several neurological conditions, the first generation [^11^C]PK11195 PET tracer has been investigated but appeared limited by poor pharmacokinetics and pharmacodynamics (low signal-to-noise ratio and unspecific binding). The second- and third-generation tracers have shown superior imaging properties, while they are not yet fully characterized [34]. The use of TSPO PET tracers has been corroborated in several glioma models and showed to (i) improve tumor and immune cell detection, (ii) provide complementary information to [^18^F]FET PET and MR imaging, (iii) be a suitable biomarker for glioma growth and immune cell infiltration, and (iv) define glioma heterogeneity in combination with other imaging modalities [33,35,36,37,38,39].

#### 3.3.1. Detecting Areas beyond FET Imaging and MRI

A human pilot study in patients with glioblastoma using the [^18^F]GE-180 tracer indicated that TSPO signal was found in areas beyond contrast-enhanced MR regions, with significant tumor-to-background contrast [38]. Although [^18^F]GE-180 has been questioned for its ability to cross the intact BBB, [^18^F]GE-180 uptake seems to be independent of BBB breakdown [33,39], and, therefore, could highlight a region of inflammation independently of dysfunctional BBB. In the same study, some contrast-enhanced areas showed lower tracer binding compared to non-contrasted regions, pointing out tissue heterogeneity and regions of distinctive cell composition within the tumor mass [38]. The PET parameters included mean background uptake (SUV_BG_), maximal tumor-to-background ratio (TBR_max_) and PET volume using different thresholds (SUV_BG_ × 1.6, 1.8 and 2.0).

The same group recently published a voxel-based analysis of [^18^F]GE-180, [^18^F]FET and CE-MRI VOIs in new diagnosed gliomas patients [39]. They compartmentalized the ipsilateral hemisphere into relative contrast-enhancement (rCE)^+^/rCE^−^, TBR_GE-180_^+^/TBR_GE-180_^−^ and TBR_FET_^+^/ TBR_FET_^−^ regions. Kaiser and colleagues reported poor correlation between rCE^+^ and TBR_FET_^+^ (*r* = 0.3, *p* < 0.001) or TBR_GE-180_^+^ (*r* = 0.3, *p* < 0.001) areas, while a stronger correlation was obtained between both PET signals (*r* = 0.8, *p* < 0.001). However, individual regression between both tracer uptake highlighted a considerable diversity in their reciprocal relationship: in overlapping areas with significant [^18^F]FET PET signal, some patients showed higher [^18^F]GE-180 signal compared to other patients showing milder tracer uptake, ultimately indicating a higher level of immune infiltration and a potentially worse outcome.

Additionally, a high fraction of rCE-negative area was positive for [^18^F]GE-180 (46 ± 27%) and [^18^F]FET PET (32 ± 18%), as previously reported [38]. It would be of high interest to track the evolution of those areas and assess their prognosis value on tumor progression, as suggested by Jensen and colleagues [41].

#### 3.3.2. TSPO Imaging of Tumor Progression

Jensen et al. (2015) reported the temporal change of [^18^F]FET and [^123^I]CLINDE tracer uptake [41]. At the baseline scan, the percentage of overlap between the CE region and [^18^F]FET VOI was larger (79–93%) than with [^123^I]CLINDE SPECT VOI (15–30%) while the percentage of overlap between the two VOI tracers was quite limited, reinforcing that the combination of [^18^F]FET, TSPO and MR imaging allows detecting the tumor mass beyond CE-MRI at baseline scan. Furthermore, follow-up change in gadolinium-CE VOI overlapped to a greater extent with baseline [^123^I]CLINDE VOI than [^18^F]FET VOI, indicating that TSPO SPECT could be more representative of progressive tumor areas than [^18^F]FET [41].

#### 3.3.3. Tumor Classification

Additional characterization indicated that [^18^F]GE-180 PET uptake was associated with the histological WHO grade, with the highest uptake values observed in WHO grade IV glioblastomas while all TSPO-negative cases were WHO grade II gliomas. Along the same lines, Su et al., indicated that the number of TSPO-positive neoplastic cells increased with glioma grade, with the highest TSPO expression level detected in confirmed glioblastoma patients [33]. These observations were in line with the current hypotheses: (i) a higher level of TSPO expression is observed in a more aggressive tumor, with the highest expression detected in (grade IV) GBM, and (ii) the level of TSPO expression correlates with the proliferative and apoptotic indices and a poorer prognosis [32].

Interestingly, the number of (Iba-1^+^) microglia/macrophages increased with tumor grade while only a few cells expressed TSPO. Accordingly, Su and colleagues reported the modelling of [^11^C]PK11195 tracer uptake in glioblastoma patients and identified different tracer uptake patterns between low-grade astrocytoma and oligodendroglioma [33], allowing patients stratification at an early stage based on tracer kinetics. Further identification of the TSPO cellular sources indicated that TSPO expression was mostly found in neoplastic cells in both histotypes of glioma, with only a small contribution from microglial cells. Therefore, differences in tracer dynamics may be explained by other factors beyond microglia/macrophage activity, such as polymorphisms, tracer delivery, tissue perfusion or heterogeneity. Overall, the results supported the suitability of TSPO imaging to stratify patients into low- and high-grade glioma expressing different levels of TSPO and its potential to detect progressive low-grade gliomas into GBM [33,40].

Additionally, a direct comparison of [^18^F]GE-180 and [^18^F]FET uptake parameters in HGG patients indicated that *IDH*-wt gliomas showed significantly elevated [^18^F]GE-180 uptake compared to *IDH*-mutant gliomas [42], while [^18^F]FET uptake did not differ with the *IDH*-mutational status, therefore, reinforcing the suitability of TSPO PET in tumor classification.

#### 3.3.4. Detecting Early Infiltration

The combination of diffusion-weighted imaging (DWI), in particular kurtosis, and [^18^F]DPA-714 PET has demonstrated a superior potential for early visualization of glioma growth and tumor infiltration than the clinical standard T_2_w-MR and [^18^F]FET PET imaging in a mouse model of P3 human GBM [36]: TSPO PET imaging allowed visualization of glioma infiltration into the contralateral hemisphere 2 weeks earlier than [^18^F]FET PET imaging. Additionally, kurtosis values were significantly increased in the glioma-bearing hemisphere at week 5 post-implantation, while T_2_w-MR based edema could only be detected from week 9 post-implantation, suggesting DWI to be more sensitive to detect early differences between the ipsilateral and contralateral hemispheres. The authors concluded on the suitability of TSPO PET imaging to detect early immune cell infiltration, together with diffusion imaging.

#### 3.3.5. Glioma-Associated Inflammation

TSPO PET imaging has been used to visualize and quantify glioma-associated neuroinflammation. However, it remains unclear to which extent the TSPO expression reflects tumor tissue or glioma-associated immune cells. In preclinical and clinical studies, TSPO expression was mostly restricted to neoplastic cells with only a small contribution of GAMs and newly formed endothelial cells [33]. The level of TSPO expression by the different glioma cell lines was addressed in vitro. Winkeler et al. indicated that rat glioma cell lines, including 9L and C6, expressed a significantly higher level of TSPO compared to GL261 and that the same glioma cell lines implanted in different strains could lead to differential PET tracer dynamics [29].

On the one hand, Zinnhardt and colleagues supported the use of TSPO PET to determine the degree of immunosuppressive myeloid cell infiltration and, therefore, its use as a prognostic imaging biomarker for mechanisms of resistance in the context of the therapeutic modulation of the immunosuppressive TME (Figure 2) [35]. TSPO was widely expressed by numerous tumor-associated HLA-DR^+^ (human leukocyte antigen D-associated) and Iba-1^+^ (microglia/macrophages) myeloid cells in a high-grade glioma patient and to a lesser extent in a low-grade glioma patient. Multiparametric flow cytometry indicated that the largest portion of GAMs was composed of CD45^med^ CD14^+++^ MDSCs and CD45^high^ CD14^+++^ GAMM, and a lower percentage of CD45^dim^ CD14^+^ microglia. Interestingly, TSPO was strongly upregulated in HLA-DR^+^ MDSCs and HLA-DR^+^ GAMM, which also co-expressed a high level of PD-L1 [35], suggesting an immunosuppressive activity. Overall, the authors supported [^18^F]DPA-714 PET imaging as an imaging readout for the degree of immunosuppressive myeloid cell infiltration.

On the other hand, Pannell et al. investigated whether TSPO upregulation in astrocytes and microglia/macrophages was restricted to a specific phenotype [44]. TSPO overexpression was observed both in vitro and in vivo after injection of TNF-inducing adenovirus while no change was observed after IL-4 stimulation, correlating with [^18^F]DPA-714 PET signal. Therefore, the authors concluded that TSPO overexpression could be induced by a pro-inflammatory microenvironment.

#### 3.3.6. Therapy Readout

The suitability of combining amino acid with TSPO PET imaging to track therapy response after TMZ chemotherapy has been addressed in NMRI^nu/nu^ mice orthotopically implanted with Gli36dEGFR cells [26] (Figure 3). The authors indicated that TMZ treatment induced a decrease in unique [^18^F]FET VOI while the exclusive area of [^18^F]DPA-714 VOI was increased post-treatment. Ex vivo characterizations, in line with the PET imaging data, showed a higher number of TSPO^+^ Iba-1^+^ and TSPO^+^ GFAP^+^ cells in TMZ-treated mice compared to control mice, indicating GAMs infiltration and increased astrocytic reactivity after treatment. The preclinical data support the use of TSPO PET imaging and the further assessment of exclusive areas of [^18^F]DPA-714 tracer uptake to detect areas of GAMs infiltration after TMZ.

Subsequently, Foray et al. employed a colony-stimulating factor 1 receptor (CSF-1R) inhibitor to efficiently deplete GAMs within the TME in an orthotopic syngeneic GL261 mouse model and assessed therapy response using [^18^F]FET and [^18^F]DPA-714 PET imaging at days 7, 14 and 21 post-injection (Figure 4) [37]. The authors observed that [^18^F]DPA-714 PET VOI was not significantly changed with GAMs depletion while the progression of glioma-associated inflammation was slowed down following inhibitor withdrawal. 

The same tracer dynamics were observed for tumor-based [^18^F]FET VOI under CSF-1R inhibition and withdrawal. The authors indicated that the TSPO PET volume increase under CSF-1R inhibition-induced GAMs depletion might result from the increase in TSPO^+^ tumor cell proliferation, as indicated by the simultaneously increased in [^18^F]FET PET volume.

The authors hypothesized that the increased peripheral immune cell infiltration triggered by GAMs depletion could promote tumor cell proliferation. Interestingly, inhibitor withdrawal may reprogram both GAMs and MDSCs toward an anti-tumorigenic phenotype, in line with the reduction in [^18^F]DPA-714 PET volume. Overall, these preclinical data supported that [^18^F]DPA-714 PET tracers may highlight an immunosuppressive TME.

#### 3.3.7. Other Features

Overall, considering the role of TSPO in the regulation of GBM development, the interpretation of the TSPO PET signal is still challenged by the time-dependent cellular sources and functions of TSPO [44]. One recent study by Fu et al. (2020) indicated that TSPO appears as a key regulator of glioma growth and especially angiogenesis through the regulation of mitochondrial oxidative phosphorylation and glycolysis [30]. TSPO-deficient GL261 glioma mouse model showed high glioma proliferation and hypoxia-induced angiogenesis, ultimately showing bigger tumor mass and extensive hemorrhagic areas. Therefore, the interpretation of PET signals using molecular imaging remains challenging due to the complex functionalities of TSPO (metabolism, angiogenesis, inflammation, etc.).

Alongside, recent investigations debate the suitability of [^18^F]GE-180 as a PET tracer to visualize and quantify TSPO expression in the brain. Back-translation into a GL261 glioma mouse model supported the use of [^18^F]GE-180 to longitudinally track TSPO expression in experimental glioma and, therefore, its applicability as a diagnostic tool in patients [45]. Clinical studies indicated that [^18^F]GE-180 uptake parameters, including median background activity and TBR_max_, are insensitive to polymorphism [38,39,42], which represents an important step forward in the field of human TSPO PET imaging. In a first pilot study using a third-generation TSPO PET tracer, [^18^F]GE-180 PET provided a high tumor-to-background ratio (TBR) in untreated and recurrent glioma [38]. Results indicated that tracer uptake characteristics did not differ significantly in primary compared to recurrent tumors (TBR_max_ = 7.31 vs. 5.86) and the PET-based glioma volume was not significantly different in primary compared to recurrent gliomas or in high affinity compared to medium affinity binders. However, the specificity and sensitivity of [^18^F]GE-180 PET tracers remain to be evaluated.

Concerning the extended investigation of TSPO functionality in glioblastoma, preclinical PET imaging studies have assessed the spatiotemporal relationship between TSPO and other pro- or anti-inflammatory markers to delineate functionally distinct tumor areas.

#### 3.3.8. TSPO and Matrix Metalloproteinases

Glioblastoma is characterized by aggressive growth and high tissue invasiveness. In this context, matrix metalloproteinases (MMPs) have been linked to increased cell proliferation, tumor invasion, migration and poor prognosis in glioma patients [31,46]. MMPs affect the neuroinflammatory milieu by modulating the expression and activity of chemokines, inflammatory cytokines, growth factors and by affecting cellular migration [47], including microglia-mediated glioma invasion [48]. Therefore, a high level of MMPs seems indicative of enhanced intracerebral invasion and neovascularization [49].

MMP-2 was significantly elevated in LGG and remained elevated in confirmed glioblastoma, while MMP-9 was better correlated with glioma grade. These findings supported the development of MMPs PET imaging tracers in low- and high-grade gliomas. [^18^F]BR-351 tracer has been reported to efficiently target MMPs, showing a higher affinity for activated MMP-9 and MMP-2 compared to other MMPs [50].

As an example, Zinnhardt et al. investigated the spatiotemporal relationship between TSPO expression and MMPs in a mouse model of human patterns of glioma pathogenesis, hypothesizing that both markers may be found at sites of glioma infiltration [51]. Using a dedicated volumetric approach on a multi-tracer and multimodal imaging dataset, each imaging biomarker delineated distinct areas of the heterogeneous glioma tissue. Interestingly, compartments of exclusive [^18^F]DPA-714 (4%) or [^18^F]BR-351 (11%) VOIs along the tumor rim could be identified. Only a small fraction of the [^18^F]DPA-714 VOI (14%) and [^18^F]BR-351 VOI (11%) was not overlapping with [^18^F]FET VOI. According to the authors, a small part of the [^18^F]BR-351-derived volume that was not detected by [^18^F]FET might hint toward regions of glioma invasion. Accordingly, GAMs were found to express MMP-9 and spatially overlapped with [^18^F]BR-351 PET signal. Overall, multi-tracer and multimodal molecular imaging approaches may allow us to gain important insights into glioma-associated inflammation and differentiate between subpopulations of functionally distinct immune cells.

## 4. Conclusions

The combination of different imaging modalities represents a suitable approach to unravel tissue heterogeneity. Given the limitations of MR and amino acid PET imaging, additional TSPO PET imaging seems beneficial to highlight metabolically active regions beyond tumor cells that support therapy resistance. Clinical studies agree on the suitability of quantitative FET and TSPO PET imaging for glioma grading and categorization, which ultimately may help in planning individualized strategies for brain tumor therapy. Additionally, combining amino acid and TSPO imaging represents an interesting way to discriminate glioma cells from glioma-associated myeloid cells. However, it remains unclear if the TSPO signals in glioma-associated myeloid cells could be associated with anti-tumorigenic or immunosuppressive phenotype/functions since their activity is hindered by glioma cells in many studies. Therefore, current research should be supported by in-depth characterization in preclinical glioma models as a back-translation approach or by the combination of different biological markers to characterize the aggressive tumor mass and TSPO.

## Figures and Tables

**Figure 1 cancers-14-03139-f001:**
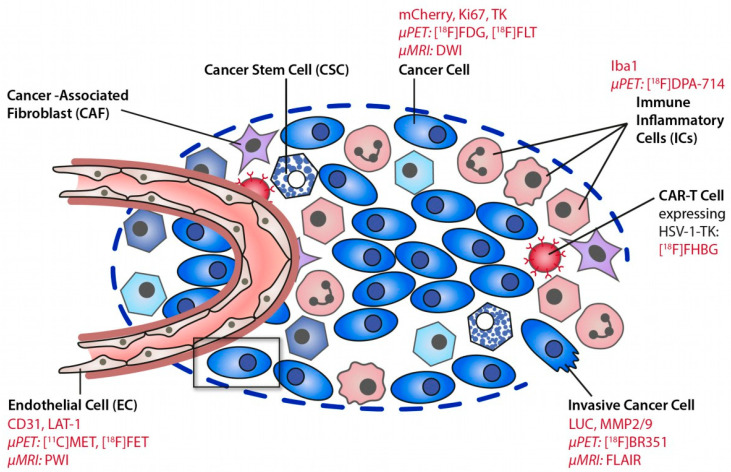
Deciphering the glioma microenvironment using molecular imaging. Different cell types of the glioma microenvironment can be targeted by MR and PET imaging. Reproduced with permission from Jacobs AH et al., *Molecular Imaging*, published by Elsevier, 2021 [10].

**Figure 2 cancers-14-03139-f002:**
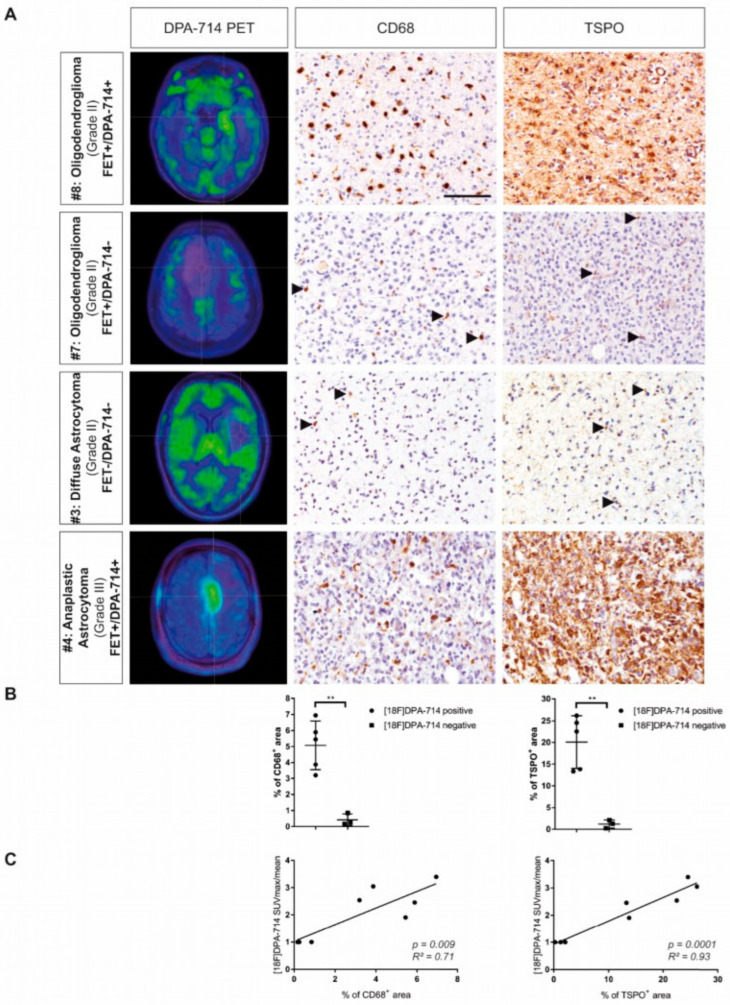
Cross-correlation of CD68^+^ cell with TSPO PET signal. (**A**) Patients with higher [^18^F]DPA-714 uptake displayed increased infiltration of CD68^+^ myeloid cells and extensive TSPO expression, while patients without [^18^F]DPA-714 uptake ([^18^F]DPA-714-negative) show only minor infiltration of CD68^+^ myeloid cells and only single cells expressing TSPO. (**B**) CD68 and TSPO immunoreactivity was increased in [^18^F]DPA-714-positive patients compared with [^18^F]DPA-714-negative patients (** *p* < 0.01). (**C**) The area of CD68 and TSPO staining correlated positively with the maximum [^18^F]DPA-714 uptake ratios. Modified and reproduced with permission from Zinnhardt et al., *Neuro-Oncology*, published by Oxford University Press, 2020 [35].

**Figure 3 cancers-14-03139-f003:**
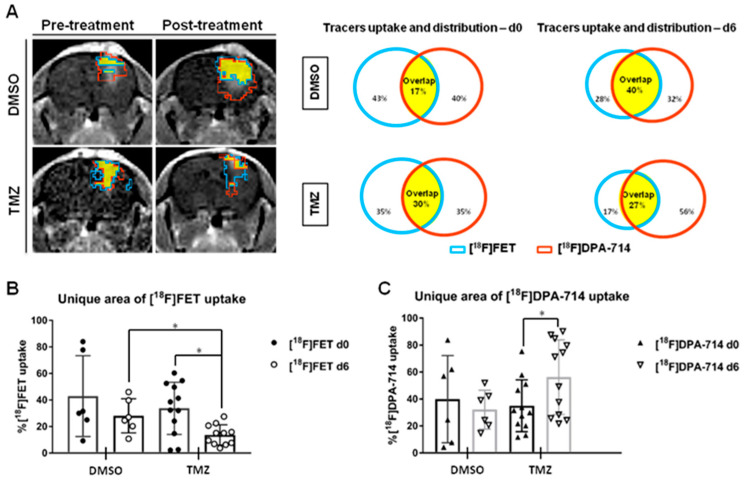
A volumetric analysis defines the distribution of tracers in the TME and highlights specific therapy-induced alterations in the uptake of individual tracers. (**A**) Representative CE-T_1_w images and single tracers VOI pre-and post-treatment in dimethyl sulfoxide (DMSO)- and temozolomide (TMZ)-treated groups. [^18^F]FET VOI (blue), [^18^F]DPA-714 VOI (red) and overlapping area (yellow) are schematically represented. (**B**) The unique area of [^18^F]FET and (**C**) [^18^F]DPA 714 tracer uptake volumes in DMSO- and TMZ-treated groups. The TMZ-treated group showed a decrease in exclusive [^18^F]FET VOI, in line with the anti-proliferative effect of TMZ while the unique [^18^F]DPA-714 VOI increased, triggered by increased immune cell infiltration in the TME after TMZ (* *p* < 0.05). Reproduced with permission from Foray et al., *Theranostics*, 2021 [26].

**Figure 4 cancers-14-03139-f004:**
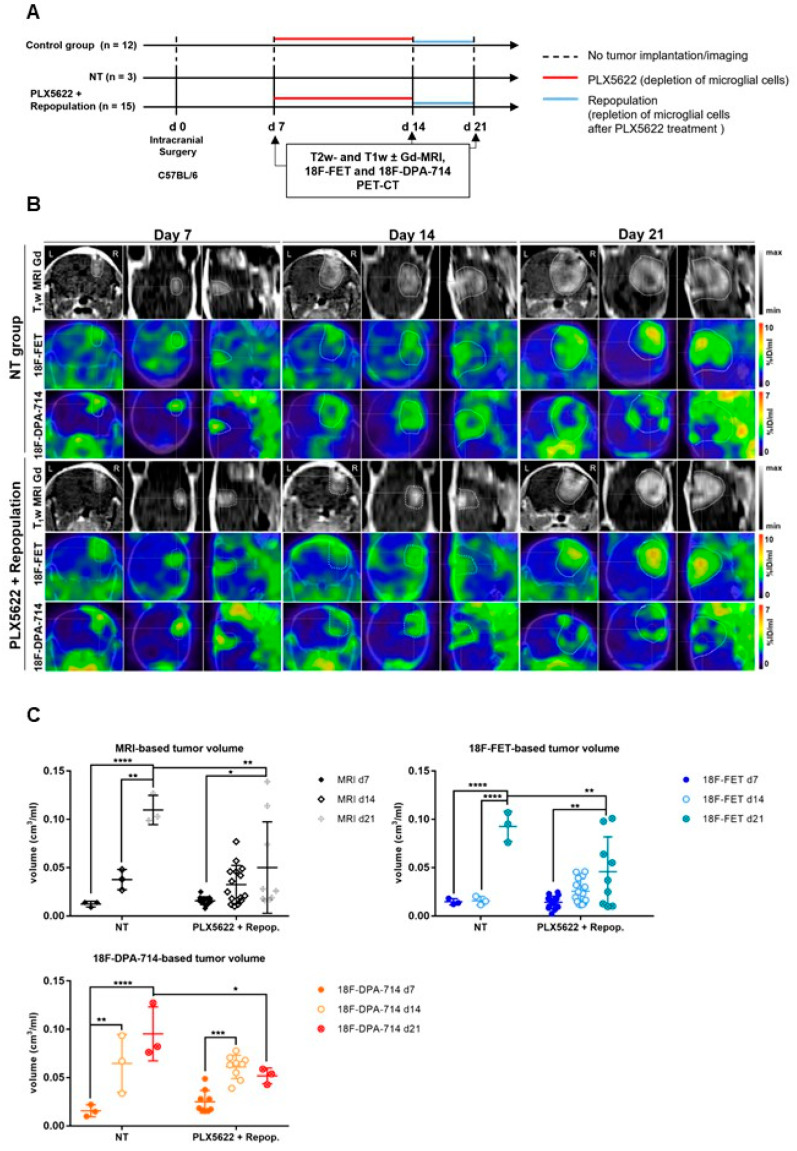
Monitoring glioma immunotherapy-induced changes after GAMs depletion and repopulation using multimodal PET/MRI. (**A**) Experimental workflow. (**B**) CE-T1w-MR and PET images for [^18^F]FET and [^18^F]DPA-714 of non-treated (NT) and PLX5622-treated animals, pre-, post-treatment, and after GAMs repopulation. (**C**) Volumetric analysis of CE-T1w- MR-, [^18^F]FET- and [^18^F]DPA-714-derived TV in NT, PLX5622-treated and repopulated groups. Repopulation (d14–d21) significantly reduced the progression of [^18^F]FET and [^18^F]DPA-714-based volume expansion while no significant difference was observed under CSF-1R inhibition-induced GAMs depletion (d7–d14). *n* = 3 NT; *n* = 15 PLX5622+repop. * *p* ≤ 0.05; ** *p* ≤ 0.01; *** *p* ≤ 0.001; **** *p* ≤ 0.0001. This research was originally published in *JNM*. Foray et al. Interrogating glioma-associated microglia/macrophage dynamics under CSF-1R therapy with multi-tracer in vivo PET/MR imaging. *J. Nucl. Med.* 2022; in press [37].

**Table 1 cancers-14-03139-t001:** Pros and Cons of each imaging modalities in the assessment of gliomas.

Modality	Pros	Cons
**MRI**	High resolution, mostly non-invasive, clinical availability	Primarily structural information
T_1_w ± CE	Tumor size and location, indicative for disrupted BBB, edema formation, hemorrhage, necrosis	Contrast dependent on a disrupted BBB, pseudoprogression, pseudoresponse
T_2_w		
FLAIR		
Diffusion	Indicative of early changes in tumor density, differentiation between GBM from lymphoma, narrowing the differential diagnosis, treatment planning	High variability
Perfusion	Neovascularization, differentiation of pseudoprogression from tumor progression	Signal quantification
**PET**	Functional/metabolic activity, high sensitivity, quantifiable	Low resolution, radiotracer production
Amino acid	Indicative of metabolic active tumor tissue, discriminate between glioma recurrence and treatment-induced changes	
TSPO	Indicative of neoplastic cells and GAMs, associated with tumor infiltration and an immunosuppressive TME	Not exclusive to neoplastic cells or GAMs
Matrix	Indicative of enhanced intracerebral invasion, neovascularization	No clinical use
metalloproteinases		

**Table 2 cancers-14-03139-t002:** Notable clinical studies in glioma patients reporting TSPO PET imaging.

Tracers	Results	Main Conclusion	Ref.
[^11^C]PK11195	*n* = 23 glioma patients, 10 healthy volunteers Three different regional kinetics were observed in individual tumors TACs: grey matter-like kinetics, white matter-like kinetics and mixed kinetics. Kinetics differed between LG astrocytoma and oligodendroglioma, independent of the tumor grade. The number of TSPO^+^ tumor cells increased with tumor grade. Only a minority of microglial cells and newly formed vessels showed TSPO expression.	Tracer kinetics in gliomas could potentially discriminate between LG astrocytomas and oligodendrogliomas	[33]
[^18^F]DPA-714 [^18^F]FET	*n* = 9, including 4 LGG and 5 HGG. Both [^18^F]FET and [^18^F]DPA-714 uptake patterns showed partial overlap with FLAIR hyperintensities. LGG patients were classified into [^18^F]FET-positive/[^18^F]DPA-714-positive and [^18^F]FET-positive/[^18^F]DPA-714-negative subgroups while all HGG patients were [^18^F]FET-positive/[^18^F]DPA-714-positive. In patients with positive uptake for both tracers, the mean percentage of overlap was 24.56%. Positive correlation between [^18^F]DPA-714 uptake and the number of (CD68^+^) GAMs (*r* = 0.84, *p* = 0.009). TSPO is strongly upregulated in HLA-DR^+^ GAMs, including HLA-DR^+^ TAMs and HLA-DR^+^ MDSCs.	[^18^F]DPA-714 may detect the glioma-associated immunosuppressive TME	[35]
[^18^F]GE-180	*n* = 10 GBM, 1 AA (confirmed *IDH*-wt glioma) All gliomas showed positive [^18^F]GE-180 uptake with high T/B contrast (median SUV_BG_: 0.47 (0.37–0.93), TBR_max_: 6.61 (3.88–9.07)). [^18^F]GE-180 uptake could be found even in areas without contrast enhancement on MR images.	First [^18^F]GE-180 imaging in patients	[38]
[^18^F]GE180 [^18^F]FET	*n* = 34 newly diagnosed glioma, including 30/34 WHO grade IV and no mutational *IDH1/2* gene. Poor correlation between rCE and PET signals was observed but a strong correlation between PET signals. More than 50% of the patients showed a large distance between rCE and TBR_GE-180_ or TBR_FET_ hottest spots. In most patients, a large proportion of voxels without increased rCE (73 ± 17%) was identified, of which a high fraction was positive in [^18^F]GE-180 (46 ± 27%) and [^18^F]FET PET (32 ± 18%), showing a large variance in TBR values for both PET signals.	Amino acid and TSPO PET imaging combined with MRI allow the depiction of tumor heterogeneity	[39]
[^11^C]PK11195	*n* = 22 (13 astrocytomas, 9 oligogendrogliomas). BP_ND_ of [^11^C]PK11195, corrected for local blood volume, in HG glioma was significantly higher than in LG astrocytoma (*p* = 0.007) and oligodendroglioma (*p* = 0.05). TSPO in gliomas was mostly expressed by neoplastic cells, correlating with BP_ND_ in tumor. GAMs accounted for 7.5–44.4% of the total cell density, with only 16.9% of GAMs expressing TSPO. TSPO expression in GAMs did not correlate with BP_ND_.	Tracer kinetics predominantly reflect TSPO^+^ glioma cells	[40]
[^123^I]CLINDE [^18^F]FET	*n* = 3 GBM patients (grade IV) The percentage of overlap between [^18^F]FET and [^123^I]CLINDE VOIs was variable (12–42%). VOIs of increased gadolinium-enhanced (Gd-CE) at baseline overlapped to a greater extent with baseline [^18^F]FET while Gd-CE VOIs at follow-up overlapped to a greater extent with baseline [^123^I]CLINDE VOIs.	TSPO PET at baseline may predict tumor progression at follow-up.	[41]
[^18^F]GE-180 [^18^F]FET	*n* = 20 HGG (9 *IDH*-wt, 11 *IDH*-mutant), including *n* = 8 newly diagnosed and *n* = 12 recurrent gliomas. *IDH*-wt gliomas showed a higher median TBR_max_ in [^18^F]GE-180 PET compared to *IDH*-mutant gliomas (median: 5.44 vs. 3.97), without reaching significance (*p* = 0.08). No difference in [^18^F]GE-180 TBR_max_ or BTV_GE-180_ was observed between newly diagnosed and recurrent HGG. The spatial correlation between BTV_GE-180_ and BTV_FET_ was only moderate, independently of the *IDH* mutation.	[^18^F]GE-180 may be susceptible to the *IDH*-mutational status	[42]
[^11^C]PBR28 [^11^C]MET	*n* = 5 patients with intracranial metastatic lesions. [^11^C]MET was accurate for detecting tumor regrowth in 7/7 brain metastases, whereas [^11^C]PBR28 was only accurate in 3/7 lesions.	[^11^C]PBR28 is not reliable to detect radiation necrosis	[43]

TAC: time–activity curve; LG: low grade; TSPO: translocator protein; BP: binding potential; HG: high grade; GAMs: glioma-associated myeloid cells; GBM: glioblastoma; VOI: volume-of-interest; AA: anaplastic astrocytoma; *IDH:* isocitrate dehydrogenase; wt: wild-type; T/B: tumor-to-background; SUV: standardized uptake value; TBRmax: tumor-to-background ratio; BTV: biological tumor volumes; rCE: relative contrast enhancement; HLA-DR: human leukocyte antigen D related; CD68: cluster differentiation 68; TME: tumor microenvironment.

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
