# Peer review of "In Vivo Quantitative Imaging of Glioma Heterogeneity Employing Positron Emission Tomography"

_cancers, 2022, doi:10.3390/cancers14133139_

Round 1
Reviewer 1 Report
The authors described the combination of various imaging modalities to detect heterogeneity in gliomas. Using conventional imaging (MRI) and amino acid tracer PET, in combination with a novel PET imaging method (TSPO PET) may help characterizing gliomas prior to surgery or during follow up.
The topic addressed in this article is important and interesting. The authors use extensive amount of literature to review the subject. The references are valid and relevant. The authors do not jump to conlcusions not supported by scientific evidence. The language is fine, the article is easy to read. Chapters help the reader in easier understanding. The figures are easy to interpret with the supporting description.
Author Response
We would like to thank the reviewer for the positive feedback.
Reviewer 2 Report
The literature survey by Andreas H Jacobs et al. on In vivo quantitative imaging of glioma heterogeneity employing positron emission tomography, the manuscript is organised well and could be of an interest to a wide range of readers. However, there are a few issues that need to be resolved before suggesting for the publication.
- The overall English of this review paper needs to be improved.
- Could the author elaborate more on the challenges the current imaging modalities in the abstract?
- Could the author rearrange the table 1 based on the results and novelty of each method rather than refereeing to the authors and publication?
- Could the authors include a table compering pros and cons of each imaging methods?
Author Response
The literature survey by Andreas H Jacobs et al. on In vivo quantitative imaging of glioma heterogeneity employing positron emission tomography, the manuscript is organised well and could be of an interest to a wide range of readers. However, there are a few issues that need to be resolved before suggesting for the publication.
We would like to thank the reviewer for the positive feedback.
- The overall English of this review paper needs to be improved.
As suggested, the manuscript has undergone extensive English revisions.
- Could the author elaborate more on the challenges the current imaging modalities in the abstract?
As suggested, we elaborated on the next challenge in the field of glioma imaging as follows (page 1, lines 30-36): “Magnetic resonance imaging (MRI) and amino acid positron emission tomography (PET) are clinically established imaging methods informing on tumor size, localization and secondary phenomena but remain quite limited in defining tumor heterogeneity, a key feature of glioma resistance mechanisms. The combination of different imaging modalities improved the in vivo characterization of the tumor mass by defining functionally distinct tissues probably linked to tumor regression, progression and infiltration. In-depth image validation on tracer specificity, biological function and quantification is critical for clinical decision making.”
- Could the author rearrange the table 1 based on the results and novelty of each method rather than refereeing to the authors and publication?
Table 1 is Table 2 in the revised manuscript (line 232).
As suggested, the above-mentioned table has been modified as follows:
- Authors and publications were deleted and replaced by the reference number, as required.
- We added a column “Main conclusion” where we highlighted the contribution of each paper to the field.
- Could the authors include a table compering pros and cons of each imaging methods?
As requested, we included an additional table (Table 1, line 114) with the pro and cons of each imaging modality.